# MATE: Multimodal Time Series Forecasting via Adaptive Modality Fusion and Timestamp-Augmented Expert Modeling

## Abstract

Time series forecasting is crucial for applications such as weather prediction, stock analysis, and power grid management. While early studies focus on modeling temporal patterns, recent studies explore multimodal approaches. However, most of these studies remain limited to converting time series into other modalities—such as images or text—without truly integrating external information. In this study, we propose MATE, a unified plug-in multimodal framework that enhances forecasting by integrating auxiliary modalities. It features a MULTI-MODAL ADAPTIVE FUSION mechanism, which dynamically selects informative modalities and routes temporal data to corresponding modality-aware experts, and a TIMESTAMP-AUGMENTED EXPERT module that treats timestamps as an independent modality to improve temporal structure awareness. Empirically, MATE achieves up to 10.32% and 12.54% improvement in MAE over unimodal and multimodal models. Extensive ablation studies assess the contribution of each modality to forecasting accuracy. The implementation code is publicly available at https://anonymous.4open.science/r/MATE-1C4D.

## 1 Introduction

Time series forecasting is a key aspect of many real-world applications, such as climate forecasting Nguyen et al. (2024), power grid management Kardakos et al. (2013), and stock analysis Koa et al. (2024). Most existing methods primarily rely on mining temporal patterns from historical observations to forecast future values. While deep learning methods excel at this Li et al. (2023); Zhang & Yan (2023); Liu et al. (2023b), they are approaching a performance barrier Song et al. (2024); Miller et al. (2024). One key limitation is their exclusive dependence on historical data, which, while capturing temporal patterns such as trends and seasonality, often omits external contextual information that governs more subtle temporal changes. For instance, in climate forecasting, temporal dynamics are closely tied to geographic and environmental conditions, where features like elevation significantly influence both the amplitude and direction of changes. These observations highlight the need for integrating additional informative modalities into forecasting models, marking a paradigm shift from single-modal to multimodal forecasting.

Current multimodal methods are generally either text-based Liu et al. (2025a;b); Chang et al. (2023); Zhou et al. (2023); Jin et al. (2023) or vision-based Li et al. (2020); Sood et al. (2021); Zhong et al. (2025); Chen et al. (2024a). However, these studies are limited in that they rely on modalities that are constructed from the time series, rather than on truly external information. For instance, some approaches embed time series into prompts for Large Language Models (LLMs) Liu et al. (2025a); Jin et al. (2023), while others convert them into images for visual models Zhong et al. (2025); Chen et al. (2024a). Motivated by their limitations, we aim to explore the integration of genuinely independent multimodal information into time series forecasting. Unlike time series data, many external modalities remain static and do not evolve. Naively fusing them with temporal patterns risks obscuring valuable sequential information. In addition, different modalities encode different characteristics, such as spatial structures in images, semantic contexts in text, and periodicities in timestamps, posing additional integration challenges. Based on these observations, this study addresses three research questions: **1)** How can multimodal data be integrated to enhance forecasting

accuracy? **2)** Which modalities contribute most to accuracy improvements? **3)** How can timestamps, as inherent temporal indicators, be effectively leveraged to improve forecasting accuracy?

To address these challenges, we propose MATE (**M**ultimodal **A**daptive fusion and **T**imestamp-augmented **E**xpert), a unified plug-in multimodal framework that leverages multimodal information to enhance time series forecasting. The core idea of MATE is to incorporate diverse modalities through adaptive selection and expert-guided processing. Specifically, dedicated encoders are first employed to extract modality-specific features. Beyond conventional fusion strategies, MATE employs a MULTIMODAL ADAPTIVE FUSION mechanism, where temporal features dynamically select the most informative modalities and route themselves to appropriate experts. Motivated by varying contributions of auxiliary modalities (see Figure 2), we design a routing mechanism to select the most informative ones per time series. To prevent information loss from directly merging dynamic temporal features with static multimodal features, and to address the distributional heterogeneity across nodes (i.e., spatial locations where several observations are collected), we adopt a Mixture-of-Experts (MoE)-like architecture. In this setup, multimodal features are fed to a router that directs them to specialized experts per node, enabling flexible and targeted processing. Furthermore, to exploit the temporal priors embedded in timestamps, we propose a TIMESTAMP-AUGMENTED EXPERT module that predicts future values from timestamps and integrates them with the backbone of time series forecasting for final prediction. We report on extensive experiments conducted on multiple benchmark datasets and state-of-the-art (SOTA) time series backbones to validate the effectiveness of MATE. We also report on comprehensive ablation studies that offer insight into the contribution of each modality and evaluate the proposed adaptive design. Our main contributions are summarized as follows:

- We propose MATE, a unified plug-in multimodal framework for time series forecasting, which can be flexibly integrated into typical backbone models.

- We introduce two key components: a MULTIMODAL ADAPTIVE FUSION mechanism that dynamically selects and fuses informative modalities, and a TIMESTAMP-AUGMENTED EXPERT module that explicitly leverages temporal priors from timestamps—an often underutilized yet readily available signal in time series data.

- We conduct extensive experiments on multiple benchmark datasets with representative backbone models to validate the effectiveness of MATE. Ablation studies further quantify each modality's contribution, demonstrating the proposed framework's interpretability.

## 2 RELATED WORK

**Time Series Forecasting.** A range of deep-learning based time series forecasting methods exist. Surveys summarize common architectures and hybrid designs that combine statistical and neural models Lim & Zohren (2021). Convolutional networks have been shown to outperform recurrent models at sequential tasks Bai et al. (2018). Unsupervised approaches learn transferable time series representations without labels Franceschi et al. (2019). Transformer-based models have advanced long-term forecasting by improving efficiency and capturing long-range dependencies Zhou et al. (2021); Wu et al. (2021); Zhou et al. (2022). However, these methods are all limited to using only time series data. To enhance time series forecasting with multimodal information, we adopt five representative backbones—TimesNet Wu et al. (2022), FEDformer Zhou et al. (2022), PatchTST Nie et al. (2022), TimeMixer Wang et al. (2024c), and Koopa Liu et al. (2023b)—which are built on fundamentally different paradigms: convolution, attention, and Koopman operators, respectively.

**Timestamp-augmented Time Series Analysis.** Timestamps contain rich global seasonality and period information, which can offer valuable guidance for time series forecasting. Traditional Transformer-based models primarily focus on local temporal patterns, often ignoring global timestamp signals Zhou et al. (2021); Wu et al. (2021); Zhou et al. (2022); Wu et al. (2022). Recent studies have explored explicit timestamp modeling to improve forecasting robustness, particularly under noise and non-stationarity Wang et al. (2024a). Moreover, the permutation-invariant nature of self-attention has prompted new architectures that better preserve temporal order and incorporate timestamp priors Zeng et al. (2023); Liu et al. (2023a). Recognizing the importance of timestamps, our study introduces timestamp priors to guide forecasting, and it enhances node-specific modeling capabilities via a lightweight decoder to handle distributional differences across nodes.

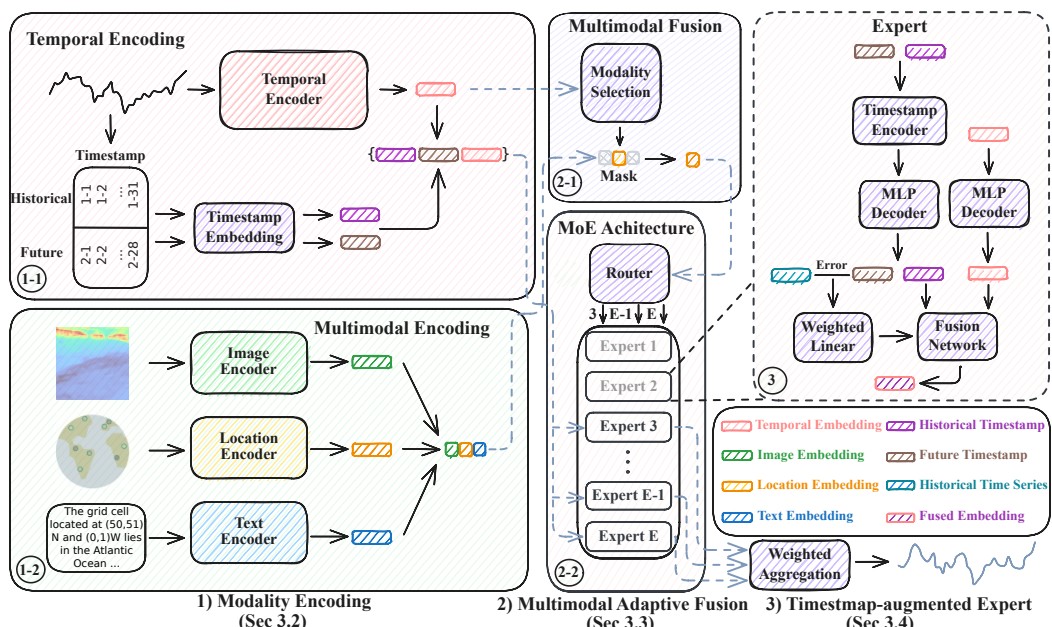

Figure 1: Overview of MATE. **1-1**: Backbone network for time series encoding and timestamp embedding; **1-2**: Multimodal input encoding; **2-1**: Selection and fusion of multimodal features; **2-2**: MoE architecture that routes and assigns features to appropriate experts; **3**: Timestamp-augmented Expert that incorporates temporal priors to improve forecasting accuracy.

**Multimodal Time Series Analysis.** Recent studies leverage large language models (LLMs) for time series analysis Xu et al. (2025). They treat time series as token sequences and adapt pre-trained language models through fine-tuning or reprogramming Gruver et al. (2023); Liu et al. (2025b); Chang et al. (2023); Zhou et al. (2023); Jin et al. (2023); Liu et al. (2024a). Unified modeling approaches have also been proposed that enable cross-domain time series forecasting by designing flexible frameworks that adapt to heterogeneous data distributions Liu et al. (2024b). In parallel, visual-based approaches recast time series forecasting as image modeling, using recurrence plots or leveraging vision foundation models pre-trained on natural images Li et al. (2020); Sood et al. (2021); Chen et al. (2024a). Additionally, general-purpose models have emerged to capture multi-scale temporal and frequency patterns for broad applicability Wang et al. (2024b). While prior methods adapt models from other modalities, they remain constrained by unimodal data. In contrast, our approach integrates truly multimodal sources to enrich forecasting and improve accuracy.

## 3 METHODOLOGY

**Problem Definition and Notation.** We consider the task of multimodal time series forecasting, where the objective is to predict future values by leveraging both historical time series data and auxiliary multimodal information. Formally, given a historical input sequence $X \in \mathbb{R}^{T_x \times N \times 1}$, the goal is to forecast future values $Y \in \mathbb{R}^{T_y \times N \times 1}$, where $T_x$ and $T_y$ denote the lengths of the historical and prediction horizons, respectively, and $N$ is the number of nodes (or sensors), each producing a univariate time series. Note that the notation table can be found in Appendix A.

In addition to $X$ and $Y$, we are also given their associated timestamps: $X_{tf} \in \mathbb{R}^{T_x \times C_{tf}}$ and $Y_{tf} \in \mathbb{R}^{T_y \times C_{tf}}$, where $C_{tf}$ is the dimensionality of the timestamp features.

Furthermore, a set of $m$ auxiliary modalities is available, denoted by $\mathcal{M} = \{M_1, M_2, \ldots, M_m\}$, where each modality $M_i \in \mathbb{R}^N$ provides static or context-specific information related to the $N$ nodes (e.g., image, text, or location).

The goal is to learn a function $\mathbf{F} : (X, \mathcal{M}, X_{tf}, Y_{tf}) \rightarrow \hat{Y}$ that maps the historical time series, auxiliary modalities, and timestamp features to future predictions $\hat{Y} \in \mathbb{R}^{T_y \times N \times 1}$.

### 3.1 OVERVIEW

As shown in Figure 1, MATE consists of three key components. The first component, MODALITY ENCODING, employs a set of modality-specific encoders to extract representations from each auxiliary modality. The second component, MULTIMODAL ADAPTIVE FUSION, adopts an MoE-like architecture that adaptively selects informative modalities and fuses the outputs of selected experts. The third component, TIMESTAMP-AUGMENTED EXPERT, incorporates both historical and future timestamps to provide temporal priors that guide future value prediction. Notably, MATE is designed as a *plug-and-play*, model-agnostic framework that can be integrated into typical backbone architectures.

### 3.2 MODALITY ENCODING

Before modality fusion, each modality is individually encoded using appropriate encoders. The available multimodal data include images $IG \in \mathbb{R}^{\mathtt{N} \times 3 \times \mathtt{H} \times \mathtt{W}}$, text descriptions $TX \in \mathbb{R}^{\mathtt{N} \times \mathtt{L}}$, and locations $LC \in \mathbb{R}^{\mathtt{N} \times 2}$, representing static attributes associated with each node. Here, $\mathtt{H}$ and $\mathtt{W}$ represent the height and width of images, $\mathtt{L}$ denotes the length of a padded text sentence, and the two location dimensions correspond to latitude and longitude. Specifically, these modalities are VGG (Earth Vertical Gravity Gradient) images, LLM-generated text, and geographic coordinates, collected per grid cell on Earth Chen et al. (2024b).

To encode the multimodal inputs, we adopt modality-specific encoders tailored to each data type. Specifically, we employ SatCLIP Klemmer et al. (2025), denoted as $\mathrm{F}_{LI}(\cdot)$, to jointly encode location ($LC$) and image ($IG$) data, aligning geographic coordinates with satellite imagery. The lightweight version of GPT-2 Radford et al. (2019) denoted as $\mathrm{F}_{TX}(\cdot)$ is used to encode text ($TX$) data. Time series data is encoded using a forecasting backbone such as TimesNet Wu et al. (2022), represented by $\mathrm{F}_T(\cdot)$. Our framework is model-agnostic and can be flexibly integrated with various backbone architectures. The overall encoding process can be expressed as follows:

$$\boldsymbol{H}_{LC}, \boldsymbol{H}_{IG} = \mathrm{F}_{LI}(LC, IG), \boldsymbol{H}_{TX} = \mathrm{F}_{TX}(TX), \boldsymbol{H}_{TS} = \mathrm{F}_T(\boldsymbol{X}), \tag{1}$$

where $\boldsymbol{H}_{LC} \in \mathbb{R}^{\mathtt{N} \times \mathtt{C_m}}$, $\boldsymbol{H}_{TX} \in \mathbb{R}^{\mathtt{N} \times \mathtt{C_m}}$ and $\boldsymbol{H}_{IG} \in \mathbb{R}^{\mathtt{N} \times \mathtt{C_m}}$, and $\boldsymbol{H}_{TS} \in \mathbb{R}^{\mathtt{N} \times \mathtt{T_y}}$, $\mathtt{T_y}$ corresponds to the prediction horizon. These prepared features are employed directly in the next step.

### 3.3 MULTIMODAL ADAPTIVE FUSION

An essential challenge in multimodal learning is how to fuse heterogeneous modalities effectively. In our setting, this challenge is amplified by the dynamic nature of time series, which fundamentally differs from static modalities such as text and location. Time series encode temporal dependencies, while the auxiliary modalities do not. Moreover, time series from different nodes often exhibit distinct distributions, necessitating node-specific processing. Simple fusion operations, like addition or concatenation, are typically inadequate for aligning and integrating such multimodal features.

To address this, we propose an MoE-like architecture. Unlike traditional MoE modules that operate at the token level, our design routes temporal features to specialized experts on a per-node basis. The fusion module consists of two key components: (1) a standard *router*, which assigns temporal features to the most suitable experts, and (2) a set of *experts*, which perform specialized processing. We employ multimodal features to guide temporal features to K specific experts. We thus feed multimodal features into the router to generate a probability table $\boldsymbol{G}$ that reflects the assignment of temporal features to specific experts for processing. Here, we define table $\boldsymbol{G}$ as follows:

$$\boldsymbol{G} = \mathrm{Softmax}(\mathrm{Gate}(\boldsymbol{H}_{mm})) \in \mathbb{R}^{\mathtt{N} \times \mathtt{E}}, \tag{2}$$

where $\mathrm{Gate}(\cdot)$ is a learnable linear projection that maps the feature dimension to the number of experts $\mathtt{E}$, and $\boldsymbol{H}_{mm}$ is the fused multimodal representation (generated in Eq. 7).

For each node, the top-K experts are selected according to the highest entries in table $\boldsymbol{G}$:

$$\boldsymbol{V}_{top_K}, \boldsymbol{I}_{top_K} = \mathrm{top_K}(G) \in \mathbb{N}^{\mathtt{N} \times \mathtt{K}}. \tag{3}$$

Then the temporal features are passed to the selected experts and gathered via weighted aggregation based on $\boldsymbol{I}_{top_K}$ and $\boldsymbol{G}$:

$$\boldsymbol{H}_{EX} = \mathrm{Select}(\mathrm{Expert}(\boldsymbol{H}_{TS}, \boldsymbol{I}_{top_K})) \in \mathbb{R}^{\mathtt{N} \times \mathtt{T_y} \times \mathtt{K}}, \tag{4}$$

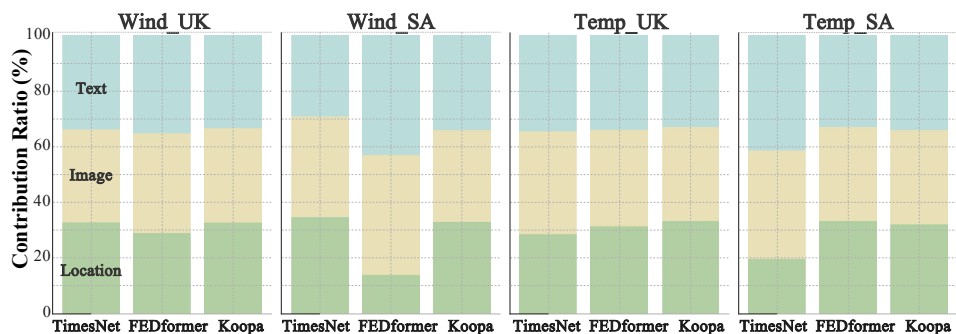

Figure 2: Single modality contribution analysis across different datasets and methods.

where Expert denotes TIMESTAMP-AUGMENTED EXPERT and Select gathers the outputs from experts. The final step uses the weights of $\boldsymbol{V}_{top_K}$ to aggregate the outputs of the K Experts:

$$\boldsymbol{H}_{pred} = \text{Aggregation}(\boldsymbol{H}_{EX}, \boldsymbol{V}_{top_K}) \in \mathbb{R}^{\mathbb{N} \times \mathtt{T_y}}, \tag{5}$$

where $\boldsymbol{H}_{pred}$ denotes the final prediction from the full model and $\mathtt{T_y}$ denotes the prediction horizon.

To fuse features from different modalities, we conduct experiments to evaluate the individual contribution of each modality. In these experiments, we employ only one modality from $[\boldsymbol{H}_{LC}, \boldsymbol{H}_{IG}, \boldsymbol{H}_{TX}]$ as $\boldsymbol{H}_{mm}$ to guide the selection of experts. As shown in Figure 2, in time series forecasting, the contribution ratio is computed by measuring the individual performance gain obtained by adding each modality to the unimodal baseline, then normalizing the gains across all modalities. Specifically, for TimesNet, the text modality contributes more to **Wind_UK** than location and image, while image contributes more to **Wind_SA** than location and text. These findings indicate that an adaptive selection mechanism is needed that can identify the most relevant modality for a given forecasting scenario. Note that we also allow combinations of modalities to better exploit complementary information.

Due to the varying contributions of auxiliary modalities observed in Figure 2, we design a modality selection mechanism that is guided by temporal features. Specifically, we adopt $\boldsymbol{H}_{TS}$ to generate a 3-element modality mask as follows:

$$\boldsymbol{Mask} = (\text{Sigmoid}(\text{Linear}(\text{Mean}(\boldsymbol{H}_{TS}, dim = 0))) > \tau) \in [0, 1]^3, \tag{6}$$

where $\text{Mean}(\cdot)$ aggregates features across all nodes to produce a shared modality mask, promoting stable training. The $\text{Linear}(\cdot)$ layer outputs a 3-element vector, corresponding to the three auxiliary modalities. The Sigmoid function ensures independent activation probabilities for each modality, and the threshold $\tau$ is applied to filter out less informative modalities. The selected modalities are then combined according to the resulting mask as follows:

$$\boldsymbol{H}_{mm} = \boldsymbol{H}_{LC} \cdot \boldsymbol{Mask}[0] + \boldsymbol{H}_{IG} \cdot \boldsymbol{Mask}[1] + \boldsymbol{H}_{TX} \cdot \boldsymbol{Mask}[2], \tag{7}$$

where $\cdot$ denotes scalar multiplication, applying each mask entry to its corresponding modality feature. The fused embedding $\boldsymbol{H}_{mm}$ thus encodes discriminative node-level semantics for expert routing.

### 3.4 TIMESTAMP-AUGMENTED EXPERT

As previously discussed, time series from different nodes generally exhibit different behaviors due to factors like geographic variation. To address this, we design a lightweight, node-specific decoder within each expert to process the temporal features. This decoder is implemented as follows:

$$\boldsymbol{H}_{PF} = \text{Linear}_3(\text{ReLU}(\text{Linear}_2(\text{Linear}_1(\boldsymbol{H}_{TS})))) \in \mathbb{R}^{\mathbb{N} \times \mathtt{T_y}}, \tag{8}$$

where $\text{Linear}_1(\cdot)$ and $\text{Linear}_2(\cdot)$ project the time series features to a higher-dimensional space, followed by $\text{Linear}_3(\cdot)$, which maps them back to the original dimensionality of $\mathbb{N} \times \mathtt{T_y}$.

Inspired by Wang et al. (2024a), we leverage timestamps as a source of prior knowledge in time series forecasting, particularly to capture periodic structures. Timestamps provide global temporal cues often not explicitly encoded in raw time series data. Since time series values are not directly correlated with their corresponding timestamps, we perform the forecasting based on timestamps

independently and then fuse the result with the output of the backbone model. For simplicity, this fusion mechanism is embedded into each expert.

We propose an encoder-decoder architecture in which a Transformer-based encoder encodes timestamp features, and a lightweight decoder—defined in Eq. 8—maps the encoded representations to time series values. Specifically, both historical and future timestamps, $\boldsymbol{X}_{tf}$ and $\boldsymbol{Y}_{tf}$, are transformed into their respective predictions $\hat{\boldsymbol{X}}$ and $\hat{\boldsymbol{Y}}$. This process is formulated as follows:

$$\hat{\boldsymbol{X}}_{hidden} = \text{Linear}_2(\text{Transformer}(\text{Linear}_1(\boldsymbol{X}_{tf}))), \tag{9}$$

where $\text{Linear}_1(\cdot)$ projects the timestamp features into the hidden space, and $\text{Linear}_2$ maps the encoded output to dimensionality $\texttt{N}$, producing a node-level representation. Then decoding through Eq. 8, we obtain $\hat{\boldsymbol{X}}$ and $\hat{\boldsymbol{Y}}$ for the next step.

Specifically, $\hat{\boldsymbol{X}}$ measures the prediction error from the historical side $\boldsymbol{X}$, and $\hat{\boldsymbol{Y}}$ is combined with the backbone prediction following the above error. We define this function as follows:

$$\boldsymbol{W} = \text{Linear}(\boldsymbol{X} - \hat{\boldsymbol{X}}) \in \mathbb{R}^{\texttt{N} \times 2}, \tag{10}$$

where $\boldsymbol{W}$ denotes the weights for aggregating $\hat{\boldsymbol{Y}}$ and $\boldsymbol{H}_{PF}$ through weighted summation. The forecasting result is then computed as follows:

$$\boldsymbol{Y}_{pred} = \boldsymbol{W}[:, 0] \odot \hat{\boldsymbol{Y}} + \boldsymbol{W}[:, 1] \odot \boldsymbol{H}_{PF} \in \mathbb{R}^{\texttt{N} \times \texttt{T}_{\texttt{y}}}, \tag{11}$$

where $\odot$ denotes element-wise multiplication and $\boldsymbol{Y}_{pred}$ is the prediction from one Expert in Eq. 4.

## 4 EXPERIMENTS

### 4.1 EXPERIMENTAL SETUP

**Datasets.** We evaluate our method on four real-world climate datasets: **Wind_SA**, **Wind_UK**, **Temp_SA**, and **Temp_UK**, covering wind and temperature measurements across two regions. All datasets are sourced from **Terra** Chen et al. (2024b), which links time series with spatial images and text descriptions. Additional details are provided in Appendix B.

**Baselines.** We evaluate our method on five representative forecasting backbones: TimesNetWu et al. (2022) (convolution-based), FEDformerZhou et al. (2022), PatchTSTNie et al. (2022), TimeMixerWang et al. (2024c) (attention-based), and KoopaLiu et al. (2023b) (physics-inspired). MATE serves as a plug-in module and can be integrated into these models. We also compare with multimodal baselines, including Time-VLMZhong et al. (2025), GLAFFWang et al. (2024a), CALFLiu et al. (2025b), and GPT4TS Zhou et al. (2023), which incorporate image or text features derived from time series. All experiments are run over 3 times in D.4 and conducted on a server with two NVIDIA A100-40GB GPUs and two Intel Xeon Gold 5218 CPUs (2.30GHz).

**Implementation Details.** We follow **Terra** Chen et al. (2024b) for forecasting settings and use **BasicTS** Shao et al. (2024) as the codebase. Data is split in a 6:2:2 ratio for training, validation, and testing. Models take the past 30 days as input and predict over 7, 15, and 30-day horizons, evaluated by MAE, RMSE, and MAPE. The time complexity and efficiency are in Appendix C.3 and D.5.

**LC** and **IG** are encoded using SatCLIP Klemmer et al. (2025) (trained from scratch), and **TX** with a 4-layer GPT-2 Radford et al. (2019). Time-VLM is re-implemented from its original description. Our fusion module uses 12 experts, selecting 3 per input, with each expert containing a two-layer Transformer. Models are trained for up to 60 epochs using Adam and MultiStepLR, with a batch size of 64 and learning rates between 0.0005 and 0.002. Additional details are in Appendix C.

### 4.2 FORECASTING RESULT ANALYSIS

Table 1 shows that MATE consistently improves forecasting across all datasets, models, and metrics. The results on PatchTST and TimeMixer are in Appendix D.1. On **Temp_SA** and **Temp_UK**, the gains are most pronounced—for example, Koopa sees an 18.9% reduction in MAE and a 13.9% drop in RMSE. These datasets contain rich contextual inputs, such as location descriptions and

Table 1: Comparison of forecasting performance of classical time series forecasting methods *with and without* the proposed MATE module on four datasets.

| Methods | Koopa | | | +MATE | | | TimesNet | | | +MATE | | | FEDformer | | | +MATE | | |
|---|---|---|---|---|---|---|---|---|---|---|---|---|---|---|---|---|---|---|
| Metric | MAE | RMSE | MAPE | MAE | RMSE | MAPE | MAE | RMSE | MAPE | MAE | RMSE | MAPE | MAE | RMSE | MAPE | MAE | RMSE | MAPE |
| **Wind_SA** 7 | 1.216 | 1.609 | 0.284 | **1.166** | **1.546** | **0.271** | 1.216 | 1.609 | 0.285 | **1.174** | **1.553** | **0.275** | 1.213 | 1.602 | 0.283 | **1.164** | **1.545** | **0.271** |
| 15 | 1.234 | 1.632 | 0.287 | **1.167** | **1.552** | **0.268** | 1.231 | 1.625 | 0.287 | **1.176** | **1.554** | **0.275** | 1.229 | 1.622 | 0.287 | **1.165** | **1.540** | **0.274** |
| 30 | 1.269 | 1.680 | 0.295 | **1.166** | **1.554** | **0.267** | 1.258 | 1.661 | 0.292 | **1.179** | **1.558** | **0.278** | 1.251 | 1.650 | 0.292 | **1.163** | **1.544** | **0.272** |
| Avg | 1.233 | 1.633 | 0.286 | **1.160** | **1.541** | **0.268** | 1.228 | 1.623 | 0.286 | **1.171** | **1.550** | **0.273** | 1.225 | 1.619 | 0.286 | **1.156** | **1.534** | **0.271** |
| **Wind_UK** 7 | 2.480 | 3.068 | 0.295 | **2.417** | **2.981** | **0.290** | 2.480 | 3.073 | 0.295 | **2.423** | **2.986** | **0.290** | 2.486 | 3.078 | 0.293 | **2.421** | **2.987** | **0.287** |
| 15 | 2.468 | 3.054 | 0.293 | **2.414** | **2.978** | **0.290** | 2.467 | 3.054 | 0.295 | **2.415** | **2.978** | **0.288** | 2.470 | 3.057 | 0.294 | **2.408** | **2.973** | **0.285** |
| 30 | 2.529 | 3.126 | 0.300 | **2.426** | **2.991** | **0.291** | 2.521 | 3.120 | 0.300 | **2.422** | **2.983** | **0.290** | 2.528 | 3.129 | 0.299 | **2.427** | **2.998** | **0.289** |
| Avg | 2.485 | 3.074 | 0.295 | **2.412** | **2.976** | **0.288** | 2.481 | 3.072 | 0.295 | **2.407** | **2.969** | **0.287** | 2.478 | 3.069 | 0.294 | **2.406** | **2.970** | **0.286** |
| **Temp_SA** 7 | 2.034 | 2.782 | 0.100 | **1.804** | **2.482** | **0.089** | 1.887 | 2.606 | 0.093 | **1.806** | **2.490** | **0.091** | 1.826 | 2.509 | 0.090 | **1.755** | **2.455** | **0.085** |
| 15 | 2.272 | 3.089 | 0.112 | **1.854** | **2.549** | **0.090** | 2.000 | 2.716 | 0.099 | **1.876** | **2.559** | **0.093** | 1.892 | 2.594 | 0.092 | **1.772** | **2.440** | **0.088** |
| 30 | 2.751 | 3.683 | 0.135 | **1.915** | **2.624** | **0.092** | 2.218 | 2.922 | 0.110 | **1.892** | **2.551** | **0.093** | 1.950 | 2.634 | 0.098 | **1.788** | **2.442** | **0.090** |
| Avg | 2.253 | 3.084 | 0.111 | **1.827** | **2.511** | **0.089** | 1.958 | 2.659 | 0.096 | **1.822** | **2.491** | **0.091** | 1.836 | 2.516 | 0.090 | **1.723** | **2.393** | **0.085** |
| **Temp_UK** 7 | 1.384 | 2.194 | 3.241 | **1.274** | **2.000** | **3.010** | 1.310 | 2.068 | 3.046 | **1.273** | **2.004** | **2.976** | 1.325 | 2.062 | 3.307 | **1.266** | **1.994** | **2.823** |
| 15 | 1.469 | 2.332 | 3.471 | **1.278** | **1.994** | **3.082** | 1.315 | 2.078 | 2.994 | **1.281** | **2.014** | **2.953** | 1.326 | 2.079 | 3.296 | **1.273** | **2.012** | **2.833** |
| 30 | 1.674 | 2.630 | 4.044 | **1.332** | **2.130** | **2.990** | 1.383 | 2.240 | 3.098 | **1.333** | **2.109** | **3.066** | 1.377 | 2.192 | 3.469 | **1.309** | **2.038** | **3.099** |
| Avg | 1.479 | 2.340 | 3.502 | **1.279** | **2.014** | **2.996** | 1.315 | 2.100 | 2.994 | **1.273** | **2.007** | **2.953** | 1.319 | 2.071 | 3.313 | **1.266** | **1.982** | **2.935** |
| **Improv. %** Avg | - | - | - | 10.32 | 10.33 | 10.73 | - | - | - | 4.44 | 4.65 | 3.46 | - | - | - | 4.68 | 4.42 | 6.23 |

infrastructure images, which MATE leverages to capture underlying semantic variability. In contrast, improvements on **Wind_SA** and **Wind_UK** are smaller, likely because strong temporal regularities in wind patterns already dominate the predictive signal, limiting the added value of external modalities.

Model-wise, Koopa benefits more from MATE than TimesNet or FEDformer. As a physics-inspired model, Koopa fails to integrate contextual information. MATE compensates for this by selectively injecting relevant multimodal cues. Meanwhile, TimesNet and FEDformer already model global dependencies and time-aware patterns effectively, so the relative impact of MATE is more moderate. These results highlight MATE's strength in enhancing models that underutilize contextual data and in tasks where external modalities offer complementary information beyond temporal dynamics.

Table 2: Ablation studies on **Wind_SA**.

| Methods | Koopa | | | TimesNet | | | FEDformer | | |
|---|---|---|---|---|---|---|---|---|---|
| Metrics | MAE | RMSE | MAPE | MAE | RMSE | MAPE | MAE | RMSE | MAPE |
| TS | 1.233 | 1.633 | 0.286 | 1.231 | 1.628 | 0.286 | 1.225 | 1.619 | 0.286 |
| TS+LC | 1.203 | 1.592 | 0.279 | 1.188 | 1.578 | 0.275 | 1.204 | 1.596 | 0.279 |
| TS+IG | 1.203 | 1.589 | 0.280 | 1.179 | 1.561 | **0.273** | 1.161 | 1.540 | 0.272 |
| TS+TX | 1.202 | 1.588 | 0.280 | 1.189 | 1.572 | 0.275 | 1.161 | 1.541 | 0.272 |
| TS+LC+IG | 1.201 | 1.592 | 0.279 | 1.197 | 1.588 | 0.278 | 1.159 | 1.537 | **0.271** |
| TS+LC+TX | 1.202 | 1.589 | 0.279 | 1.195 | 1.585 | 0.277 | 1.160 | 1.540 | 0.272 |
| TS+IG+TX | 1.201 | 1.590 | 0.280 | 1.179 | 1.561 | **0.273** | 1.160 | 1.540 | 0.272 |
| TS+LC+IG+TX | 1.202 | 1.589 | 0.279 | 1.180 | 1.565 | 0.274 | 1.163 | 1.541 | 0.273 |
| TS+Adpt(LC+IG+TX) | 1.202 | 1.588 | 0.279 | 1.175 | 1.556 | 0.274 | 1.159 | 1.536 | **0.271** |
| **TS+Adpt(LC+IG+TX)+TA** | **1.160** | **1.541** | **0.268** | **1.171** | **1.550** | **0.273** | **1.156** | **1.534** | **0.271** |

### 4.3 ABLATION STUDIES

We conduct ablation studies to examine the contribution of each modality and module in our framework. Starting from the **TS** baseline (time series only), we evaluate the impact of incorporating **LC** (location), **IG** (image), and **TX** (text), both individually and in combinations (**LC+IG**, **LC+TX**, **IG+TX**, and **LC+IG+TX**). We further apply **Adpt(LC+IG+TX)** using Multimodal Adaptive Fusion to dynamically select informative modalities, and examine **TA**, which introduces timestamp priors during decoding to enhance temporal modeling.

Table 2 shows results on **Wind_SA** using the 30-day prediction average; additional results appear in Appendix D.2. The results show that incorporating external modalities consistently improves forecasting accuracy over the **TS**-only baseline. For FEDformer and TimesNet, adding **IG** notably

reduces MAE (e.g., from 1.225 to 1.161 and from 1.231 to 1.179), suggesting that visual geographic cues effectively complement these temporally dominant models by capturing location-dependent wind patterns. In contrast, Koopa benefits almost equally from **LC** and **IG**, which aligns with its spatiotemporal design and reliance on structured dynamics, indicating that coarse location information alone can already provide meaningful spatial signals for Koopman-based modeling.

Combining modalities like **LC+IG** or **LC+TX** yields modest yet consistent gains, suggesting complementary information. However, naive fusion quickly saturates—especially for Koopa—indicating potential redundancy or conflict when modalities are treated equally.

To address this, we apply **Adpt(LC+IG+TX)** for adaptive selection. This improves performance across the board, with TimesNet reaching 1.175 MAE and FEDformer 1.159, indicating that selective attention to informative modalities mitigates the noise introduced by irrelevant ones. The benefits are more pronounced in TimesNet and FEDformer, likely due to their greater flexibility in learning modality-specific dynamics compared to Koopa's fixed operator framework.

Finally, adding **TA** further improves performance by injecting temporal priors. The largest gain appears in Koopa (1.202 to 1.160 MAE), likely because it lacks strong native temporal encoding. TimesNet and FEDformer also benefit, though to a lesser extent, possibly because their architectures already leverage positional or frequency-based temporal mechanisms. These results highlight the importance of both adaptive fusion and temporal priors in enhancing multimodal forecasting.

## 4.4 COMPARISON WITH MULTIMODAL METHODS

Table 3 compares our method with recent multimodal forecasting approaches. FEDformer+MATE consistently outperforms all baselines across datasets and metrics, demonstrating robust performance gains. Compared with Time-VLM, which derives modalities from time series, and GLAFF, which emphasizes timestamp usage, our method leverages truly independent multimodal inputs for stronger complementarity. CALF and GPT4TS, though based on large models, show weaker generalization, especially on **Temp_SA**—likely due to domain mismatch and lack of targeted fusion. Notably, FEDformer+MATE improves over GPT4TS by over 12% on all metrics, validating the effectiveness of our lightweight, adaptive fusion design.

Table 3: Comparison of forecasting performance over multimodal methods on four datasets.

| Methods | | FEDformer+MATE | | | Time-VLM | | | GLAFF | | | CALF | | | GPT4TS | | |
|---|---|---|---|---|---|---|---|---|---|---|---|---|---|---|---|---|
| Metrics | | MAE | RMSE | MAPE | MAE | RMSE | MAPE | MAE | RMSE | MAPE | MAE | RMSE | MAPE | MAE | RMSE | MAPE |
| **Wind_SA** | 7 | **1.164** | **1.545** | **0.271** | 1.195 | 1.582 | 0.280 | 1.213 | 1.602 | 0.283 | 1.215 | 1.604 | 0.285 | 1.216 | 1.617 | 0.278 |
| | 15 | **1.165** | **1.540** | **0.274** | 1.195 | 1.577 | 0.283 | 1.232 | 1.624 | 0.288 | 1.232 | 1.627 | 0.287 | 1.233 | 1.626 | 0.289 |
| | 30 | **1.163** | **1.544** | **0.272** | 1.201 | 1.589 | 0.281 | 1.258 | 1.658 | 0.294 | 1.267 | 1.678 | 0.293 | 1.267 | 1.678 | 0.292 |
| | Avg | **1.156** | **1.534** | **0.271** | 1.196 | 1.581 | 0.281 | 1.226 | 1.619 | 0.286 | 1.227 | 1.624 | 0.285 | 1.227 | 1.626 | 0.284 |
| **Wind_UK** | 7 | **2.421** | **2.987** | **0.287** | 2.427 | 2.994 | 0.290 | 2.488 | 3.089 | 0.295 | 2.516 | 3.110 | 0.303 | 2.485 | 3.072 | 0.298 |
| | 15 | **2.408** | **2.973** | **0.285** | 2.427 | 2.995 | 0.290 | 2.476 | 3.070 | 0.295 | 2.482 | 3.069 | 0.297 | 2.474 | 3.061 | 0.298 |
| | 30 | **2.427** | **2.998** | **0.289** | 2.433 | 3.001 | **0.289** | 2.550 | 3.156 | 0.300 | 2.551 | 3.156 | 0.305 | 2.535 | 3.133 | 0.302 |
| | Avg | **2.406** | **2.970** | **0.286** | 2.416 | 2.983 | 0.288 | 2.489 | 3.086 | 0.294 | 2.494 | 3.086 | 0.297 | 2.483 | 3.072 | 0.296 |
| **Temp_SA** | 7 | **1.755** | **2.455** | **0.085** | 1.922 | 2.652 | 0.095 | 1.841 | 2.515 | 0.091 | 2.031 | 2.820 | 0.101 | 2.076 | 2.842 | 0.102 |
| | 15 | **1.772** | **2.440** | **0.088** | 1.940 | 2.656 | 0.097 | 1.898 | 2.582 | 0.094 | 2.269 | 3.134 | 0.112 | 2.379 | 3.181 | 0.118 |
| | 30 | **1.788** | **2.442** | **0.090** | 2.059 | 2.818 | 0.103 | 1.951 | 2.615 | 0.099 | 2.711 | 3.651 | 0.134 | 3.020 | 3.964 | 0.149 |
| | Avg | **1.723** | **2.393** | **0.085** | 1.940 | 2.669 | 0.097 | 1.840 | 2.503 | 0.092 | 2.218 | 3.086 | 0.109 | 2.363 | 3.207 | 0.117 |
| **Temp_UK** | 7 | **1.266** | **1.994** | **2.823** | 1.360 | 2.174 | 2.724 | 1.294 | 2.044 | 3.096 | 1.377 | 2.174 | 3.237 | 1.380 | 2.178 | 3.205 |
| | 15 | **1.273** | **2.012** | **2.833** | 1.382 | 2.235 | 2.941 | 1.315 | 2.098 | 3.168 | 1.471 | 2.326 | 3.590 | 1.471 | 2.312 | 3.481 |
| | 30 | **1.309** | **2.038** | 3.099 | 1.445 | 2.337 | **3.015** | 1.356 | 2.185 | 3.237 | 1.675 | 2.594 | 4.242 | 1.672 | 2.593 | 4.137 |
| | Avg | **1.266** | **1.982** | 2.935 | 1.375 | 2.206 | **2.892** | 1.302 | 2.078 | 3.147 | 1.481 | 2.341 | 3.543 | 1.475 | 2.315 | 3.542 |
| **Improv. %** | Avg | - | - | - | 5.72 | 5.98 | 3.78 | 4.54 | 4.51 | 5.58 | 11.54 | 11.77 | 11.95 | 12.54 | 12.19 | 13.11 |

## 4.5 T-SNE ANALYSIS

Figure 3 presents t-SNE visualizations of node embeddings on **Wind_UK** over Koopa; additional results are in Appendix D.3. The left plot shows embeddings from the baseline model, while the right plot includes multimodal fusion. Each color corresponds to one of 100 nodes, with 10 samples per node. In the left panel, node embeddings overlap heavily, showing limited spatial discrimination

from temporal data alone. The right panel instead exhibits clearer separation and tighter clustering, as auxiliary modalities enrich spatial semantics. This more structured latent space aligns with the observed performance gains, indicating that improved representations drive better results.

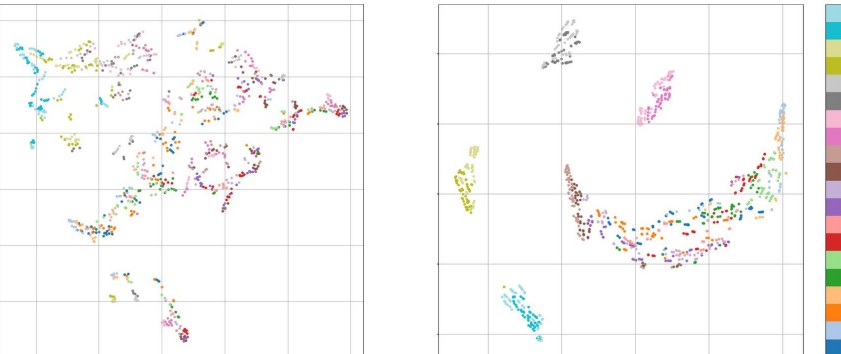

Figure 3: t-SNE visualization on **Wind_UK** over Koopa. **Left:** w/o MATE; **Right:** w/ MATE.

## 4.6 PARAMETER ANALYSIS

We assess the impact of expert selection using three configurations—2/8, 3/12, and 4/16—where K/E denotes the number of selected experts K out of a total of E. Experiments are conducted on **Wind_UK** and **Temp_UK** using TimesNet, Koopa, and FEDformer. As shown in Figure 4, performance remains largely stable across settings. TimesNet and FEDformer achieve slightly better MAE and RMSE at 3/12 on both datasets, while Koopa exhibits minimal sensitivity. These results suggest that our method is robust to the choice of expert configuration, maintaining consistent accuracy without the need for fine-grained parameter tuning.

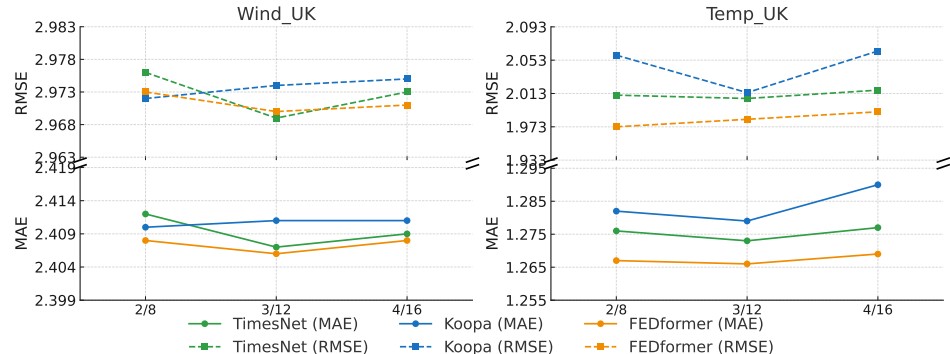

Figure 4: Parameter analysis.

## 5 CONCLUSION

In this work, we present MATE, a unified plug-in multimodal framework that enhances time series forecasting by integrating auxiliary modalities in an adaptive and timestamp-aware manner. Through MULTIMODAL ADAPTIVE FUSION and TIMESTAMP-AUGMENTED EXPERT, MATE enables modality selection, dynamic expert routing, and timestamp-aware predictions, leading to consistent improvements across diverse benchmarks. Unlike prior approaches that merely re-encode time series into other modalities, MATE effectively utilizes auxiliary modalities to enrich temporal representations while maintaining the interpretability and modularity of the forecasting pipeline.

Future work includes extending MATE to handle asynchronous or partially missing modalities, and adapting its timestamp-aware design to model hierarchical or multi-scale temporal patterns for improved robustness.

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

# A    NOTATION TABLE

| Notation | Description |
|----------|-------------|
| $X$ | Historical input |
| $Y$ | Future values |
| $X_{tf}$ | Historical timestamps |
| $Y_{tf}$ | Future timestamps |
| $\mathcal{M} = \{M_1, M_2, \ldots, M_m\}$ | Auxiliary modality set |
| $\hat{Y}$ | Future predictions |
| $IG$ | Image modality |
| $TX$ | Text modality |
| $LC$ | Location modality |
| $H_{LC}$ | Feature of location |
| $H_{IG}$ | Feature of image |
| $H_{TX}$ | Feature of text |
| $H_{TS}$ | Feature of time series |
| $Y_{pred}$ | Prediction from one expert |
| $H_{EX}$ | Predictions from selected experts |
| $H_{pred}$ | Final prediction |

Table 4: Summary of main notations used in this paper.

# B    DATASET DESCRIPTION

In this section, we provide a detailed description of the datasets used in our study. All data are derived from the **Terra** Chen et al. (2024b) and include both time series and multimodal data:

- **Time Series**: The Terra dataset provides global wind speed and temperature records from October 1996 to June 2022. In our study, we utilize data from two countries, South Africa (SA) and the UK. These regions are divided into grid cells with a resolution of 1 degree, resulting in 100 and 221 grid regions (nodes) for the SA and UK, respectively. Each region has its independent time series. The data are recorded at a daily granularity, with each series containing a single channel (wind or temperature). We evaluate forecasting performance over three prediction lengths: 7, 15, and 30 days.

- **Coordinates**: For each region in the SA and UK, we use the geographic center point as its location representation.

- **Images**: The dataset includes Earth Vertical Gravity Gradient images for each region in the UK and SA. These images reflect variations in Earth's gravity, offering insights into subsurface structures and topographical features that may influence local environmental conditions.

- **Text**: For each region, basic attributes such as climate, average elevation, land vegetation type, and country affiliation are obtained from Köppen Climate Classification Project Chen & Chen (2013). These attributes are further processed and enriched using large language models to generate standardized and comprehensive textual descriptions.

- **Timestamps**: Following Terra's standard representation, we use *day of month* and *month of year* as timestamps, as meteorological data generally lack clear periodicity.

# C    IMPLEMENTATION DETAILS

## C.1    MODEL CONFIGURATION

Table 5 summarizes the configurations of the text, image, and timestamp encoders used in our framework, including the number of layers, hidden dimensions, and other key architectural parameters.

Table 5: Model configurations for text, image, and timestamp encoders.

| Module | Parameter | Value |
|---|---|---|
| **Text Encoder (Shallow GPT2)** | Number of Transformer layers | 4 |
| | Hidden dimension | 768 |
| | Feedforward dimension | 3072 |
| | Number of attention heads | 12 |
| | Projection output dimension | 128 |
| **Image Encoder (SatCLIP)** | Input resolution | 224 |
| | Patch size | 16 |
| | Vision layers | 4 |
| | Vision width (hidden dim) | 768 |
| | Number of attention heads | 12 |
| | Output dimension | 128 |
| **Timestamp Encoder** | Number of Transformer layers | 2 |
| | Hidden dimension | 16 |
| | Feedforward dimension | 32 |
| | Number of attention heads | 1 |
| | Normalization | LayerNorm |

## C.2    METRICS

To evaluate the forecasting performance, we adopt three commonly used metrics: mean absolute error (MAE), root mean squared error (RMSE), and mean absolute percentage error (MAPE). To account for missing or invalid values (e.g., NaNs or near-zero entries), we use their masked versions, where all computations are performed only on valid entries. The corresponding formulas are defined as follows:

- **MAE (Mean Absolute Error)**:

$$\text{MAE} = \frac{1}{Z} \sum_{i=1}^{n} m_i \cdot |y_i - \hat{y}_i|, \tag{12}$$

  where $y_i$ and $\hat{y}_i$ denote the ground truth and prediction, $m_i$ is a binary mask indicating valid values, and $Z = \sum_i m_i$ is the normalization factor.

- **RMSE (Root Mean Squared Error)**:

$$\text{RMSE} = \sqrt{\frac{1}{Z} \sum_{i=1}^{n} m_i \cdot (y_i - \hat{y}_i)^2}. \tag{13}$$

- **MAPE (Mean Absolute Percentage Error)**:

$$\text{MAPE} = \frac{1}{Z} \sum_{i=1}^{n} m_i \cdot \left| \frac{y_i - \hat{y}_i}{y_i} \right| \tag{14}$$

  where we apply a mask $m_i = 0$ when $|y_i| < \epsilon$ to avoid numerical instability.

These metrics provide complementary insights: MAE evaluates average error magnitude, RMSE emphasizes large deviations, and MAPE captures relative error sensitivity. All metrics are averaged over the valid spatiotemporal elements.

## C.3    TIME COMPLEXITY

Our framework consists of a 4-layer ViT, a 4-layer GPT-2, and a MoE module with 12 experts, each being a 2-layer Transformer. The total time complexity is approximately:

$$O\left(BN_{\text{patch}}^2 H\right) + O\left(BL_{\text{txt}}^2 H\right) + O\left(NE\right) + O\left(NKBT^2 H\right) \tag{15}$$

where B is the batch size, N is the number of spatial nodes, H is the hidden dimension, T is the input sequence length, K is the Top-K selection size, and E is the number of experts.

# D    MORE RESULTS

## D.1    MORE FORECASTING RESULTS

Compared to the earlier baselines, both PatchTST and TimeMixer also benefit from MATE, though in different ways. For PatchTST, the improvements are substantial across all datasets, since PatchTST primarily relies on local temporal patch representations, it struggles to capture high-level contextual signals. MATE complements this limitation by injecting external semantic cues, leading to significant error reductions.

For TimeMixer, the gains are more moderate but still consistent. TimeMixer already models hierarchical temporal dependencies effectively, which partly explains the smaller relative improvements on **Wind_SA** and **Wind_UK**. Nonetheless, on the temperature datasets where auxiliary information is richer, MATE provides clear added value. Overall, these findings reinforce that MATE is especially effective for architectures underutilizing contextual modalities, while still offering steady improvements to stronger temporal models.

Table 6: Comparison of forecasting performance of extra baselines *with and without* the proposed MATE module on four datasets.

| Methods | PatchTST | | | +MATE | | | TimeMixer | | | +MATE | | |
|---|---|---|---|---|---|---|---|---|---|---|---|---|
| Metric | MAE | RMSE | MAPE | MAE | RMSE | MAPE | MAE | RMSE | MAPE | MAE | RMSE | MAPE |
| **Wind_SA** 7 | 1.213 | 1.602 | 0.283 | **1.174** | **1.556** | **0.273** | 1.203 | 1.590 | 0.280 | **1.183** | **1.566** | **0.275** |
| 15 | 1.230 | 1.625 | 0.286 | **1.175** | **1.559** | **0.272** | 1.216 | 1.609 | 0.282 | **1.185** | **1.572** | **0.275** |
| 30 | 1.263 | 1.671 | 0.293 | **1.176** | **1.564** | **0.273** | 1.235 | 1.634 | 0.286 | **1.189** | **1.587** | **0.273** |
| Avg | 1.224 | 1.620 | 0.285 | **1.169** | **1.552** | **0.271** | 1.208 | 1.600 | 0.280 | **1.179** | **1.567** | **0.272** |
| **Wind_UK** 7 | 2.490 | 3.078 | 0.295 | **2.418** | **2.981** | **0.290** | 2.448 | 3.018 | 0.295 | **2.430** | **2.997** | **0.288** |
| 15 | 2.477 | 3.065 | 0.294 | **2.411** | **2.974** | **0.288** | 2.433 | 3.002 | 0.293 | **2.416** | **2.979** | **0.289** |
| 30 | 2.541 | 3.140 | 0.302 | **2.427** | **2.992** | **0.289** | 2.473 | 3.048 | 0.300 | **2.439** | **3.013** | **0.295** |
| Avg | 2.487 | 3.077 | 0.295 | **2.412** | **2.974** | **0.289** | 2.443 | 3.012 | 0.294 | **2.420** | **2.986** | **0.289** |
| **Temp_SA** 7 | 2.067 | 2.825 | 0.102 | **1.807** | **2.462** | **0.090** | 2.000 | 2.762 | 0.099 | **1.845** | **2.550** | **0.091** |
| 15 | 2.356 | 3.175 | 0.116 | **1.849** | **2.498** | **0.094** | 2.192 | 3.003 | 0.109 | **1.905** | **2.612** | **0.094** |
| 30 | 2.964 | 3.908 | 0.146 | **1.884** | **2.524** | **0.096** | 2.531 | 3.451 | 0.125 | **1.951** | **2.650** | **0.095** |
| Avg | 2.339 | 3.184 | 0.115 | **1.798** | **2.442** | **0.091** | 2.153 | 2.971 | 0.107 | **1.848** | **2.543** | **0.091** |
| **Temp_UK** 7 | 1.376 | 2.177 | 3.213 | **1.281** | **2.023** | **2.817** | 1.359 | 2.181 | 2.944 | **1.280** | **2.019** | **2.962** |
| 15 | 1.475 | 2.344 | 3.404 | **1.289** | **2.045** | **2.884** | 1.440 | 2.320 | 3.079 | **1.284** | **2.027** | **2.998** |
| 30 | 1.654 | 2.583 | 4.032 | **1.339** | **2.162** | **2.943** | 1.599 | 2.578 | 3.357 | **1.338** | **2.144** | **3.023** |
| Avg | 1.471 | 2.327 | 3.480 | **1.290** | **2.053** | **2.866** | 1.440 | 2.325 | 3.090 | **1.287** | **2.038** | **2.990** |
| **Improv. %** Avg | - | - | - | 10.75 | 10.66 | 11.37 | - | - | - | 7.04 | 7.42 | 5.66 |

## D.2    ABLATION STUDY

Tables 7, 8, and 9 present the ablation results across **Wind_UK**, **Temp_SA**, and **Temp_UK**. Adding single modalities (**LC**, **IG**, or **TX**) consistently improves performance, where **IG** often brings the largest gains, particularly for temperature datasets, while **LC** benefits Koopa by providing spatial cues.

Simple modality combinations offer further but limited improvements, suggesting additive fusion alone is insufficient. Introducing **Adpt(LC+IG+TX)** enables more effective integration by dynamically selecting useful modalities, resulting in consistent performance boosts.

Finally, incorporating the **TA** brings additional gains, especially for Koopa on **Temp_SA** and **Temp_UK**. Since Koopa lacks inherent temporal encoding, **TA** notably enhances its forecasting capability. Overall, both **Adpt(LC+IG+TX)** and **TA** are critical for fully leveraging multimodal signals in time series forecasting.

Table 7: Ablation studies on **Wind_UK**.

| Methods | Koopa | | | TimesNet | | | FEDformer | | |
|---|---|---|---|---|---|---|---|---|---|
| Metrics | MAE | RMSE | MAPE | MAE | RMSE | MAPE | MAE | RMSE | MAPE |
| TS | 2.485 | 3.074 | 0.295 | 2.481 | 3.072 | 0.295 | 2.478 | 3.069 | 0.294 |
| TS+LC | 2.425 | 2.993 | 0.289 | 2.414 | 2.982 | 0.288 | 2.426 | 2.992 | 0.290 |
| TS+IG | 2.423 | 2.990 | **0.288** | 2.413 | 2.979 | **0.287** | 2.414 | 2.979 | 0.290 |
| TS+TX | 2.424 | 2.991 | 0.289 | 2.412 | 2.978 | **0.287** | 2.415 | 2.978 | 0.290 |
| TS+LC+IG | 2.422 | 2.989 | **0.288** | 2.416 | 2.985 | 0.288 | 2.411 | 2.975 | 0.288 |
| TS+LC+TX | 2.424 | 2.992 | 0.289 | 2.415 | 2.983 | **0.287** | 2.417 | 2.984 | 0.289 |
| TS+IG+TX | 2.422 | 2.989 | **0.288** | 2.415 | 2.982 | **0.287** | 2.416 | 2.983 | 0.288 |
| TS+LC+IG+TX | 2.422 | 2.991 | 0.289 | 2.416 | 2.984 | 0.288 | 2.414 | 2.980 | 0.287 |
| TS+Adpt(LC+IG+TX) | 2.422 | 2.989 | **0.288** | 2.408 | 2.973 | **0.287** | 2.408 | 2.972 | 0.287 |
| **TS+Adpt(LC+IG+TX)+TA** | **2.412** | **2.976** | **0.288** | **2.407** | **2.969** | **0.287** | **2.406** | **2.970** | **0.286** |

Table 8: Ablation studies on **Temp_SA**.

| Methods | Koopa | | | TimesNet | | | FEDformer | | |
|---|---|---|---|---|---|---|---|---|---|
| Metrics | MAE | RMSE | MAPE | MAE | RMSE | MAPE | MAE | RMSE | MAPE |
| TS | 2.253 | 3.084 | 0.111 | 1.966 | 2.676 | 0.096 | 1.836 | 2.516 | 0.090 |
| TS+LC | 1.996 | 2.775 | 0.098 | 1.911 | 2.638 | 0.093 | 1.743 | 2.392 | 0.086 |
| TS+IG | 1.983 | 2.771 | 0.098 | 1.858 | 2.533 | 0.092 | 1.742 | 2.386 | 0.086 |
| TS+TX | 1.981 | 2.774 | 0.098 | 1.851 | 2.527 | **0.091** | 1.744 | 2.383 | 0.086 |
| TS+LC+IG | 1.991 | 2.781 | 0.098 | 1.853 | 2.540 | **0.091** | 1.727 | **2.381** | **0.085** |
| TS+LC+TX | 1.991 | 2.775 | 0.098 | 1.851 | 2.536 | **0.091** | 1.740 | **2.381** | 0.086 |
| TS+IG+TX | 1.995 | 2.799 | 0.098 | 1.858 | 2.534 | 0.092 | 1.735 | 2.395 | 0.086 |
| TS+LC+IG+TX | 1.988 | 2.781 | 0.098 | 1.856 | 2.539 | 0.092 | 1.734 | 2.395 | **0.085** |
| TS+Adpt(LC+IG+TX) | 1.979 | 2.761 | 0.097 | 1.843 | 2.522 | **0.091** | 1.745 | **2.381** | 0.086 |
| **TS+Adpt(LC+IG+TX)+TA** | **1.827** | **2.511** | **0.089** | **1.822** | **2.491** | **0.091** | **1.723** | 2.393 | **0.085** |

Table 9: Ablation studies on **Temp_UK**.

| Methods | Koopa | | | TimesNet | | | FEDformer | | |
|---|---|---|---|---|---|---|---|---|---|
| Metrics | MAE | RMSE | MAPE | MAE | RMSE | MAPE | MAE | RMSE | MAPE |
| TS | 1.479 | 2.340 | 3.502 | 1.315 | 2.100 | 2.994 | 1.319 | 2.071 | 3.313 |
| TS+LC | 1.390 | 2.244 | 3.013 | 1.291 | 2.068 | 3.015 | 1.270 | 2.016 | 2.940 |
| TS+IG | 1.389 | 2.235 | 3.000 | 1.284 | 2.038 | 2.982 | 1.265 | 1.986 | **2.934** |
| TS+TX | 1.391 | 2.249 | 3.062 | 1.286 | 2.050 | **2.930** | 1.266 | 1.984 | 2.981 |
| TS+LC+IG | 1.442 | 2.327 | 3.031 | 1.287 | 2.041 | 3.018 | 1.271 | 1.998 | 2.950 |
| TS+LC+TX | 1.414 | 2.253 | 3.001 | 1.286 | 2.045 | 3.007 | 1.269 | 1.980 | 2.975 |
| TS+IG+TX | 1.390 | 2.215 | **2.996** | 1.287 | 2.041 | 3.000 | 1.268 | 1.976 | 2.977 |
| TS+LC+IG+TX | 1.392 | 2.226 | 3.031 | 1.285 | 2.038 | 3.045 | **1.264** | 1.977 | 3.017 |
| TS+Adpt(LC+IG+TX) | 1.389 | 2.215 | 3.021 | 1.283 | 2.040 | 3.001 | **1.264** | **1.968** | 2.882 |
| **TS+Adpt(LC+IG+TX)+TA** | **1.279** | **2.014** | 2.996 | **1.273** | **2.007** | 2.953 | 1.266 | 1.982 | 2.935 |

### D.3 T-SNE VISUALIZATION

To better understand the effect of our proposed MATE module, we visualize the node embeddings on **Wind_UK** using t-SNE. As shown in Figure 5 and Figure 6, the baseline models (TimesNet and FEDformer) exhibit relatively disordered or loosely clustered node distributions, indicating weaker modeling of node-level characteristics. After integrating MATE, the embeddings become noticeably more organized and structured. Specifically, nodes form clearer and more continuous trajectories in TimesNet+MATE, and exhibit tighter and more coherent clusters in FEDformer+MATE. These observations suggest that our method effectively enhances the modeling of node-specific features and global relationships, leading to more distinguishable and meaningful latent representations.

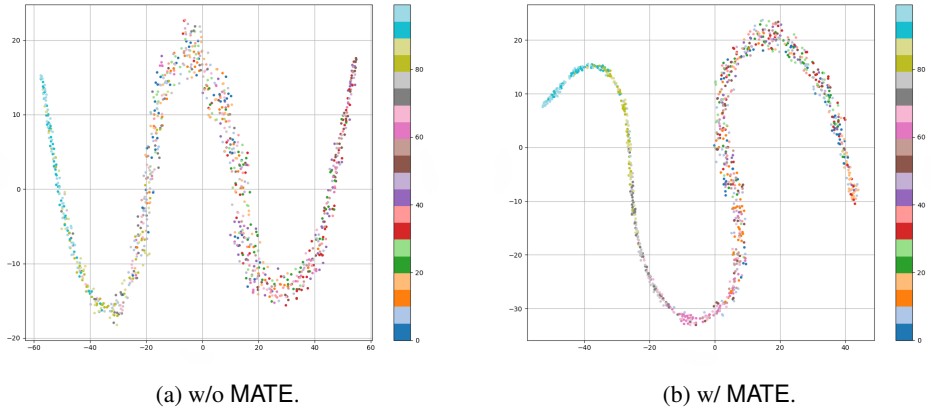

(a) w/o MATE.                                    (b) w/ MATE.

Figure 5: t-SNE on **Wind_UK** over TimesNet.

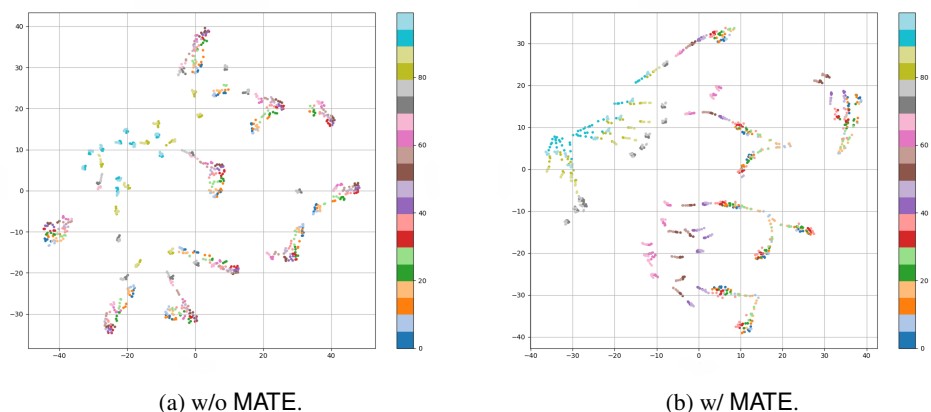

(a) w/o MATE.                                    (b) w/ MATE.

Figure 6: t-SNE on **Wind_UK** over FEDformer.

### D.4 STANDARD ERRORS

Here are all the statistical results over 3 independent runs on our four datasets. A smaller standard deviation indicates that our framework demonstrates better stability.

Table 10: Statistical results.

| Dataset | Metric | Wind_UK | Wind_SA | Temp_UK | Temp_SA |
|---|---|---|---|---|---|
| TimesNet+MATE | MAE | $2.407 \pm 0.001$ | $1.171 \pm 0.002$ | $1.273 \pm 0.002$ | $1.822 \pm 0.003$ |
| | RMSE | $2.969 \pm 0.002$ | $1.550 \pm 0.002$ | $2.007 \pm 0.003$ | $2.491 \pm 0.004$ |
| FEDformer+MATE | MAE | $2.406 \pm 0.001$ | $1.156 \pm 0.003$ | $1.266 \pm 0.002$ | $1.723 \pm 0.001$ |
| | RMSE | $2.970 \pm 0.003$ | $1.534 \pm 0.002$ | $1.982 \pm 0.003$ | $2.393 \pm 0.001$ |
| Koopa+MATE | MAE | $2.412 \pm 0.001$ | $1.160 \pm 0.002$ | $1.279 \pm 0.001$ | $1.827 \pm 0.003$ |
| | RMSE | $2.976 \pm 0.002$ | $1.541 \pm 0.001$ | $2.014 \pm 0.003$ | $2.511 \pm 0.002$ |

## D.5 EFFICIENCY COMPARISON

To compare the efficiency and model size of the MATE framework with the baselines, we record the statistics for the baselines themselves as well as when combined with MATE. The increase in model size mainly comes from the encoders of the auxiliary modalities. Similarly, the longer time per epoch is due to the encoding of multimodal data. The detailed results are as follows:

Table 11: Comparison of run time (in seconds) and data count across models.

| Model | Run Time (Wind_UK) | Run Time (Temp_UK) | Count |
|---|---|---|---|
| TimesNet | 385s | 274s | 1,830,431 |
| TimesNet+MATE | 1040s | 1840s | 108,440,608 |
| Koopa | 68s | 104s | 1,341,825 |
| Koopa+MATE | 1230s | 1640s | 107,952,002 |
| FEDformer | 130s | 402s | 14,163,659 |
| FEDformer+MATE | 1636s | 3780s | 120,790,348 |

## E THE USE OF LARGE LANGUAGE MODELS (LLMS)

We use LLMs to polish some sentences in the paper and adjust the formatting of the tables.