# OpenReview forum: "MATE: Multimodal Time Series Forecasting via Adaptive Modality Fusion and Timestamp-Augmented Expert Modeling"
_ICLR.cc/2026/Conference — Submitted to ICLR 2026_

### Official Review · Reviewer_71xB · 2025-10-30

**Soundness:** 3
**Presentation:** 2
**Contribution:** 2
**Rating:** 4
**Confidence:** 3

**Summary:**

The research paper introduces MATE, a unified plug-in multimodal framework designed to enhance time series forecasting by effectively integrating auxiliary data modalities.

The central issue the authors address is the performance limitation of existing time series forecasting methods, including advanced deep learning models, which rely almost exclusively on mining temporal patterns from historical data. This dependence omits crucial external contextual information that influences subtle temporal changes. While some recent work explores multimodal approaches, they are often limited to modalities constructed from the time series itself, such as converting data into images or text, rather than genuinely independent external information. The naive fusion of static external data with dynamic temporal patterns also risks obscuring valuable sequential information, and the inherent distributional heterogeneity across different time series nodes poses further integration challenges.

The purpose of MATE is to overcome these limitations by proposing a framework that explores the integration of genuinely independent multimodal information to enrich forecasting and improve accuracy. The authors specifically aim to answer how multimodal data can be effectively integrated, which modalities are most informative, and how timestamps can be better leveraged as temporal indicators.

Existing unimodal methods suffer from exclusive dependence on historical data, leading to the omission of essential external contextual factors, such as geographic or environmental conditions, that govern temporal changes. Current multimodal methods, despite their name, are often constrained by unimodal data as they convert the time series into another format for processing, failing to integrate truly external or independent information. Furthermore, simple data fusion techniques are inadequate for aligning heterogeneous features, particularly the static nature of auxiliary data like location and the dynamic nature of the time series, necessitating a more sophisticated, node-specific processing strategy to handle distinct data distributions. Finally, traditional Transformer models tend to focus on local temporal patterns and often ignore the rich global temporal signals embedded in timestamps.

The core idea of MATE is to incorporate diverse modalities through a strategy of adaptive selection and expert-guided processing. The framework is comprised of two primary components:

MULTIMODAL ADAPTIVE FUSION: This mechanism utilizes a Mixture-of-Experts architecture, which is MoE-like, to address distributional heterogeneity by routing temporal features to specialized experts on a per-node basis. Crucially, it employs a modality selection mechanism that is guided by temporal features to dynamically select the most informative auxiliary modalities like image, text, or location for a given time series, as their contributions vary significantly across scenarios.

TIMESTAMP-AUGMENTED EXPERT: This module treats timestamps as an independent modality. It leverages the temporal priors, such as global seasonality and period information, embedded in both historical and future timestamps. It predicts future values independently from timestamps and then aggregates this prediction with the output of the time series backbone using a weighted fusion based on the prediction error of the historical time steps.

The authors assert that this design works well because it is a plug-in, model-agnostic framework that systematically integrates truly external context and vital temporal priors while employing adaptive mechanisms to address the complexities of heterogeneous multimodal data.

The method's effectiveness is validated through extensive experiments on four real-world climate datasets: Wind SA, Wind UK, Temp SA, and Temp UK. MATE is integrated into five state-of-the-art time series backbones, including TimesNet, FEDformer, and Koopa, to demonstrate its plug-in and model-agnostic capabilities. The performance is evaluated using metrics like Mean Absolute Error, Root Mean Square Error, and Mean Absolute Percentage Error.

The experiments show that MATE consistently improves forecasting accuracy across all datasets, models, and metrics. Empirically, MATE achieves an improvement of up to ten point thirty-two percent in MAE over unimodal models and up to twelve point fifty-four percent over multimodal models.

The main contribution is the proposal of MATE as a unified plug-in multimodal framework featuring the two novel key components: the MULTIMODAL ADAPTIVE FUSION mechanism for dynamic modality selection and the TIMESTAMP-AUGMENTED EXPERT module for explicit temporal prior leveraging. Comprehensive ablation studies further provide insights into the contribution of each modality, demonstrating the framework's interpretability.

**Strengths:**

The use of a Mixture-of-Experts MoE-like architecture with per-node routing is a sound logical approach to manage the known issue of distinct data distributions across different sensor nodes and to prevent the degradation of dynamic temporal information from naive fusion with static external data.

The experimental evidence showing that the contribution of auxiliary modalities varies significantly across datasets logically necessitates and justifies the development of the adaptive selection mechanism to filter out uninformative or distracting context.

The explicit incorporation of timestamps as an independent modality addresses the logical shortcoming of many prior models that focus only on local temporal patterns while ignoring global temporal cues such as seasonality and period information.

The method solves the core problem well by successfully integrating truly external multimodal context image, location, and text, which is a vital source of information often missing from a purely historical view of time series data.

The dynamic modality selection ability of the MULTIMODAL ADAPTIVE FUSION mechanism is the key to solving the problem robustly, allowing the model to adaptively choose the most useful external context for each forecasting scenario.

A key contribution is MATE's design as a unified plug-in framework that is model-agnostic, allowing it to be flexibly integrated and enhance performance across diverse backbone models built on different paradigms like convolution or attention.

**Weaknesses:**

The paper does not provide a sufficient logical explanation for the specific choice of pre-trained encoders, such as SatCLIP and lightweight GPT-2, for encoding the auxiliary data. A more detailed rationale or a comparative analysis of different encoding strategies is needed to logically support that these choices are optimal for time series contextualization.

While the use of a lightweight node-specific decoder within the expert is mentioned, the logical rationale for the specific Multi Layer Perceptron structure with linear and ReLU layers used to process the temporal features could be more deeply explored or justified against alternative processing units.

The logical intuition behind using the historical prediction error from the timestamp expert $\text{X} - \hat{\text{X}}$ to generate the weights for fusing future predictions is not clearly explained in a compelling way. This complex weighting mechanism, which uses past performance as a gauge for future reliability, requires a more rigorous theoretical or empirical justification.

While experiments are strong on multiple backbones and use four climate datasets, the exclusive focus on wind and temperature data limits the claim of the framework's general plug-in applicability. Evaluation on datasets from vastly different domains, such as energy consumption, traffic flow, or finance, would provide more sufficient evidence for its universality.

**Questions:**

Could the authors elaborate on the specific rationale for selecting the SatCLIP and lightweight GPT-2 pre-trained models for encoding the auxiliary image, location, and text data? Were comparative experiments performed with other encoding models to confirm that these choices are optimal for generating representations that best guide the time series forecasting process?

The weighted aggregation of the timestamp-predicted future values $\hat{\text{Y}}$ and the backbone predictions $\text{H}_{PF}$ uses weights derived from the historical prediction error $\text{X} - \hat{\text{X}}$. Could the authors provide a more intuitive or theoretical explanation for why the error measured on the historical time steps serves as the optimal mechanism to determine the fusion weights for the future time steps, particularly concerning non-stationary data?

The current evaluation is restricted to climate datasets. Have the authors tested MATE on datasets from significantly different domains, such as financial, traffic, or industrial data, to confirm its claim as a truly unified plug-in framework that generalizes beyond weather forecasting?

Given the introduction of multiple components including the router and twelve experts, what is the precise increase in training time and inference latency of MATE relative to the vanilla backbone models, and are these computational costs justifiable by the performance gains for deployment in real-time or large-scale operational systems?

---

### Official Review · Reviewer_mrYV · 2025-10-30

**Soundness:** 3
**Presentation:** 3
**Contribution:** 2
**Rating:** 4
**Confidence:** 4

**Summary:**

This work proposes MATE, a multimodal framework for time-series forecasting that is model-agnostic and can be applied to any unimodal forecasting backbone. Within MATE, the authors introduce a MoE mechanism based on timestamp and temporal embeddings for modality selection. Experiments on four multimodal datasets demonstrate its effectiveness in improving forecasting accuracy compared to both unimodal baselines and existing multimodal forecasting models.

**Strengths:**

1. The proposed framework is clearly presented, easy to follow, and intuitively reasonable.

2. The use of MoE for modality selection appears to be a novel contribution of this work.

3. The evaluations demonstrate that MATE effectively improves performance on wind and temperature datasets using different unimodal forecasting backbones.

**Weaknesses:**

The experimental evaluation could be further strengthened, particularly from the following three perspectives:

1. Comparison with More Advanced Unimodal Baselines:
In Table 1, it would be beneficial to include more recent and advanced unimodal forecasting models such as iTransformer, TimeMixer, or PAttn. This would provide stronger evidence that MATE truly benefits from multimodal information. Conversely, if MATE does not show improvement on these stronger backbones, it might suggest that unimodal time-series information alone is sufficient for accurate forecasting, and the contribution of additional modalities could be limited.

2. Comparison with Stronger Multimodal Baselines:
In Table 2, the included multimodal baselines are primarily methods that reprogram time-series into other modalities, and many of them do not outperform standard time-series models like Fedformer. Incorporating more advanced multimodal forecasting approaches, such as TaTS [1] or other recent methods trained directly on multimodal datasets, would provide a fairer and more convincing comparison to highlight MATE’s advantages.

3. Evaluation on More Diverse Datasets:
Additional datasets such as PEMS or Time-MMD could be included for broader evaluation. For example, PEMS is a well-known spatio-temporal traffic dataset, and Time-MMD combines time-series data with textual information. Testing MATE on these datasets could provide two major benefits:

(1). First, these datasets are well-benchmarked, allowing for clearer comparison and better assessment of MATE’s general effectiveness.

(2). Second, since these datasets contain varying modalities (some with missing image or text components), they can help evaluate MATE’s generalization ability under incomplete modality settings, which is often more realistic in practice.
A simple way to implement this would be to introduce placeholders for missing modalities during training and inference.

[1]. Language in the Flow of Time: Time-Series-Paired Texts Weaved into a Unified Temporal Narrative.

**Questions:**

Please refer to the weakness.

---

### Official Review · Reviewer_mtH3 · 2025-10-31

**Soundness:** 2
**Presentation:** 3
**Contribution:** 2
**Rating:** 4
**Confidence:** 3

**Summary:**

MATE (Multimodal Adaptive Timestamp-aware Expert) is a pluggable multimodal framework that injects auxiliary modalities—location text (TX), remote-sensing imagery (IG), and location code (LC)—into arbitrary time-series forecasting backbones (e.g., TimesNet, FEDformer, Koopa, PatchTST, TimeMixer).

**Strengths:**

Easy to understand.

Pluggable and backbone-agnostic: works with TimesNet, FEDformer, Koopa, PatchTST, and TimeMixer.

Provides parameter sensitivity (experts/Top-K) and t-SNE visualization analyses.

Code is provided, which helps reproducibility.

**Weaknesses:**

Efficiency: Each epoch becomes much slower after adding MATE (e.g., TimesNet: 385s → 1040s/1840s; FEDformer: 130s/402s → 1636s/3780s).

Scope and generalization: All four datasets come from the Terra meteorology domain; cross-domain transfer (traffic, retail, energy) is not evaluated.

Task framing and fairness: The paper effectively does spatio-temporal forecasting on Terra but titles the work “Multimodal Time Series Forecasting,” which can be misleading. Limiting comparisons to generic time-series backbones—without including spatio-temporal or meteorology-specific models—seems unfair. Please include comprehensive comparisons to the Terra paper baselines and follow-ups such as “Expand and Compress: Exploring Tuning Principles for Continual Spatio-Temporal Graph Forecasting.”

Motivation clarity: The method tackles a spatio-temporal task but is aims to be “model-agnostic for time-series forecasting backbones.” Confused about this

**Questions:**

Check weakness

---

### Official Review · Reviewer_d4WY · 2025-10-31

**Soundness:** 3
**Presentation:** 3
**Contribution:** 2
**Rating:** 2
**Confidence:** 4

**Summary:**

MATE proposes a plug-in multimodal framework for time-series forecasting that (1) adaptively selects and fuses truly external modalities (location, satellite imagery, and LLM-generated text) via a modality-guided MoE router, and (2) adds a Timestamp-Augmented Expert that treats timestamps as an explicit modality to provide temporal priors. Integrated with common backbones (TimesNet, FEDformer, PatchTST, TimeMixer, Koopa), MATE delivers consistent gains—reporting up to ~10% MAE improvement over unimodal and ~12% over multimodal baselines on Terra climate datasets (Wind/Temp in SA/UK)—with ablations showing which modalities help most per task.

**Strengths:**

- Instead of relying only on text or time-series plot images, the method flexibly incorporates multiple, heterogeneous external modalities (e.g., spatial, environmental, or metadata cues), broadening the information surface available to the forecaster.
- The framework is easy to attach to existing TS models with minimal changes, enabling quick adoption across architectures and tasks (plug-in, backbone-agnostic design).

**Weaknesses:**

- LC/IG/TX are used only to choose experts; they never enter the prediction path. This likely under-utilizes multimodal signal and caps potential gains.
- Experiments are limited to Terra UK/SA at 1° grids with univariate targets (wind/temperature) and static auxiliaries, leaving generalization to other regions, variables, and dynamic exogenous effects uncertain.
- A node-averaged, global 3-way mask (LC/IG/TX) applies the same on/off to all nodes, reducing fine-grained, node-specific modality selection.

**Questions:**

- Where is the MoE control? Please add a head-to-head ablation of TS-only MoE (gate = (H_{\text{TS}})) versus **MM-gated MoE (gate = (H_{mm})), both **with and without Timestamp-Augmented (TA) experts. This will isolate the incremental benefit of multimodal gating and make its advantage over vanilla MoE explicit.
- I question the extent to which Earth Vertical Gravity Gradient (VGG) imagery contributes prediction-relevant signal; would a text-based representation (e.g., structured descriptors) offer comparable efficacy at substantially lower computational cost?

---

### Meta-Review · Area_Chair_9R8E · 2026-01-04

**Summary:**

This paper proposes MATE, a plug-in multimodal framework to enhance time series forecasting effectively integrating auxiliary data modalities with MoE mechanism. When integrated with common backbones, MATE shows consistent gains on datasets, reporting improvements over unimodal and multimodal baselines.
However, the paper leaves several concerns from the reviewers unresolved or unanswered. These include: a lack of ablation studies on MoE control, insufficient intuitive demonstration of the contribution of images to predictive signals, and the necessity to select other modalities instead of using high-quality text-based representation. As a plug-in multimodal framework, it should enhance prediction accuracy and generalize across multiple diverse time-series forecasting tasks, but without incurring excessive efficiency costs. Additionally, the study could incorporate the latest methods such as iTransformer and TimeMixer, and conduct evaluations on a broader range of datasets. Further investigation into the model's performance under conditions of partially missing multimodal data would help assess the generalization ability of MATE in real-world, incomplete-modality settings, which are often more common in practice.

**Reviewer Concerns:**

As the authors did not participate in the rebuttal discussion, the reviewers' concerns have not been addressed.

**Reviewer Scores:**

Given the authors' absence from the rebuttal discussion, the reviewers will maintain their original scores.

---

### Decision · Program_Chairs · 2026-01-26

Reject